

# WRF-DL v1.0: A Bridge between WRF v4.3 and Deep Learning Parameterizations and its Application to Atmospheric Radiative Transfer

Xiaohui Zhong[1], Zhijian Ma[1], Yichen Yao[1], Lifei Xu[1], Yuan Wu[1], and Zhibin Wang[1]

[1]Damo Academy, Alibaba Group, Hangzhou 311121, China

**Correspondence:** Zhibin Wang(zhibin.waz@alibaba-inc.com)

**Abstract.**

In numerical weather prediction (NWP) models, physical parameterization schemes are the most computationally expensive components, despite being greatly simplified. In the past few years, an increasing number of studies have demonstrated that deep learning (DL) parameterizations of subgrid physics have the potential to accelerate and even outperform conventional physic-based schemes. However, as the DL models are commonly implemented using the DL libraries written in Python, very few DL-based parameterizations have been successfully integrated with NWP models due to the difficulty of embedding Python functions into Fortran-based NWP models. To address this issue, we developed a coupler to allow the DL-based parameterizations to be coupled with a widely used NWP model, i.e., the Weather and Research Forecasting (WRF) model. Similar to the WRF I/O methodologies, the coupler provides the options to run the DL model inference with exclusive processors or the same processors for WRF calculations. In addition, to demonstrate the effectiveness of the coupler, the DL-based radiation emulators are trained and coupled with the WRF model successfully.

## 1 Introduction

Numerical weather prediction (NWP) models have become the most important tools for operational weather forecasting and have many applications in different domains, including energy, traffic, logistics, and planning (Coiffier, 2011; Pu and Kalnay, 2018). The physics-based parameterizations are essential in the NWP models as they approximate some processes that are either too small scale to be explicitly resolved at the model grid resolution or too complex and not fully understood. Although many simplifications are made to parameterizations to reduce the computational cost, the calculations of physical parameterizations still account for a significant portion of the total computational time of NWP models (Wang et al., 2019). Also, the parameterization schemes often contain uncertain parameters estimated from more faithful high-resolution simulations with statistical models (Stensrud, 2013). For example, the parameters in radiative transfer parameterizations can be fitted to the output of the most accurate line-by-line model (Clough et al., 2005), and parameters in cloud turbulence parameterizations can be inferred from large-eddy simulations (LES) (Mason, 1989). However, other parameters can only be learned from observations as the related governing equations are unknown (Schneider et al., 2017).



One alternative is to train deep learning (DL) models to replace the traditional physics-based parameterization schemes.
The DL-based parameterizations have the potential to outperform traditional parameterizations with higher computational efficiency. For example, the radiative transfer parameterization scheme is one of the most computationally expensive components in NWP models, and it has the longest history of developing DL-based radiation emulators. Chevallier et al. (1998, 2000) developed an NN-based longwave radiation parameterization (NeuroFlux) and has been used operationally in the European Centre for Medium-Range Weather Forecasts (ECMWF) four-dimensional variational data assimilation system. The NeuroFlux is seven times faster than the previous scheme with comparable accuracy (Janisková et al., 2002). Recently, Song and Roh (2021) developed and used the neural network (NN) based radiation emulators in the operational weather forecasting model in the Korea Meteorological Administration. They demonstrated that using NN-based emulators frequently can improve real-time weather forecasting in terms of accuracy and speed compared to the original method, which infrequently uses the original radiation parameterization.

Similarly, DL-based emulators have been developed for other parameterization schemes in NWP models. Rasp et al. (2018) successfully developed and coupled an NN-based convection parameterization into an aquaplanet general circulation model (GCM). They showed that the NN-based parameterization was able to perform multi-year simulations, of which results were close to that of the original simulations. For planetary boundary layer (PBL) parameterization, Wang et al. (2019) used the inputs and outputs from Yonsei University (YSU) PBL scheme of the Weather Research Forecast (WRF) model to develop the NN-based parameterizations. They showed that the NN-based parameterization successfully simulated the vertical profiles of velocities, temperature, and water vapor within the PBL over the entire diurnal cycle. Urban land surface models (ULSMs) are reasonably fast, but none of them is best at predicting all the main surface fluxes (Grimmond et al., 2010, 2011). One solution is to run an ensemble of ULSMs coupled to the NWP models to improve predictions, which is technically difficult to implement and would require more computational time due to running multiple ULSMs simultaneously. Meyer et al. (2022a) developed an urban NN (UNN) to emulate the ensemble mean of 22 ULSMs. They demonstrate that the WRF model coupled with the UNN is more accurate than WRF with a single well-known USLM. Grundner et al. (2021) demonstrated the potential of an NN-based cloud cover parameterization to accurately reproduce the sub-grid scale cloud fraction to improve the low-resolution climate model predictions.

In many cases, DL-based parameterization schemes are only evaluated in offline settings due to the practical challenge of coupling DL-based emulators with NWP models. However, the offline performance of DL-based emulators does not necessarily reflect their online performance when coupled with other components of NWP models. DL-based parameterizations are often trained using high-level programming languages like Python and easy-to-use DL libraries such as Pytorch (Paszke et al., 2017) and Keras (Gulli and Pal, 2017). However, the NWP codes are mainly written in Fortran, making it difficult to integrate directly with DL-based emulators. Researchers used multiple approaches to circumvent these issues. Krasnopolsky (2014) developed a single-layer NN software for developing NN-based radiation emulators in NWP models (Belochitski and Krasnopolsky, 2021; Krasnopolsky et al., 2010; Song and Roh, 2021). Some researchers wrote a Fortran module that reproduces NN architectures using matrix-matrix multiplication with the weights saved from offline training (Chantry et al., 2021; Hatfield et al., 2021; Rasp et al., 2018), which only works well for simple NN architectures such as the fully connected (FC) NNs, and becomes





increasingly difficult to implement for more complex NN structures. Ott et al. (2020) produced an open-source software library, the Fortran-Keras Bridge (FKB), which enables users to access many features of the Keras API in Fortran and implement NNs in Fortran code. However, the FKB can only support FC or dense layers NNs. Therefore, researchers cannot fully use the most advanced NN structures, such as convolution, self-attention, and variational autoencoder structures, to build powerful DL-based emulators for online applications. Wang et al. (2022) proposed an NN-GCM coupler to allow Python-based NNs to communicate with the Fortran-based GCM. Although using the coupler can make the DL-based emulators be deployed efficiently, its requirement for data transfer on a hard disk makes it hard to achieve speedup.

To address the abovementioned problems, we developed a WRF-DL coupler that allows any DL-based parameterization schemes to be coupled with the WRF model. The WRF model is selected as it is a popular open-source NWP model used by many researchers and operational organizations. Also, to demonstrate the applicability of the coupler, we train and couple the DL-based radiation emulators with the WRF model. The DL-based radiation emulators were studied most among all the physical parameterization schemes and coupled with the ECMWF Integrated Forecasting System (IFS), the National Centers for Environmental Prediction (NCEP) Climate Forecast System (CFS), and the WRF model in the previous studies (Song and Roh, 2021; Meyer et al., 2022b). Also, the rapid radiative transfer model for general circulation models (RRTMG; Iacono et al., 2008) is selected for radiation as it is widely used in weather and climate models, such as ECMWF IFS, NCEP Global Forecast System (GFS), and the China Meteorological Administration (CMA) Global/Regional Assimilation and Prediction System (GRAPES). To train the DL-based radiation emulators, the WRF model is run to generate a dataset for training and evaluation. We also illustrate the performance of the DL-based radiation emulators in the online setting. Lastly, we compare the accuracy and computational costs of the WRF simulations coupled with DL-based radiation emulators with the WRF simulations using original RRTMG schemes.

The remainder of the paper is organized as follows. Section 2 describes the original and DL-based RRTMG module. Section 3 introduces the WRF I/O quilt processors and the WRF-DL coupler. Section 4 briefly presents the WRF model setup and DL-based radiation emulators. Section 5 presents the analysis and results in online settings. Conclusions and discussions are in Section 6.

## 2 Description of Original and DL-based RRTMG model

### 2.1 Description of Original RRTMG Module

The RRTMG is developed by the Atmospheric and Environmental (AER) and comprises shortwave (SW) and longwave (LW) schemes that calculate SW and LW radiation fluxes and heating rates, respectively. Figure 1 presents the pseudo-code for the original radiation module in the WRF model. At the beginning of the radiation driver module, the input and output variables are initialized. Then, the corresponding radiation calculations are started according to the parameters (i.e., ra_lw_physics and ra_sw_physics) specified in the configuration file (i.e., namelist.input). When the RRTMG schemes are selected, the subroutine rrtmg_lwinit is called to perform calculations needed to initialize the longwave calculations. Next, the subroutines rrtmg_lwrad and rrtmg_swrad are called to perform the RRTMG SW and LW radiation calculations.



```
subroutine radiation_driver
        defining variables
        initializing data
        call data preprocessing related subroutines, e.g.: call
        radconst, cldfra, ozone or aerosol related interpolation
        lwrad_select:
        select case (config_flag%ra_lw_physics)
                case (RRTMG_LWSCHEME)
                        call rrtmg_lwinit
                        call rrtmg_lwrad
                case (other lw schemes)
        end select lwrad_select
        swrad_select:
        select case (config_flag%ra_sw_physics)
                case (RRTMG_SWSCHEME)
                        call rrtmg_swrad
                case (other sw schemes)
        end select swrad_select
end subroutine radiation_driver
```

**Figure 1.** Pseudo-code for the original RRTMG module

### 2.1.1 Description of DL-based RRTMG Module

The original radiation computations process an atmospheric column at a time and loop over all the horizontal grid points. The three-dimensional variables associated with the grid cell (indexed as $(i, j, k)$ in the WRF model output files) are indexed as $(i, j, k)$ in memory by WRF to make it more cache friendly and increase the cache hit rate. The index $i$, $j$, and $k$ represent the west-east, south-north, and vertical directions, respectively. With the DL-based RRTMG model, inference can be implemented on a batch of data, which requires the input data to be packed into batches. The input features of DL-based radiation emulators have dimensions of $W \times H$, where $W$ and $H$ represent the number of features and vertical levels, respectively. Therefore, the original three-dimensional variables are converted to be indexed as $(i, j, k)$ to match the dimensions requirement of DL-based input features (done by preprocess_layout in Figure 2). The DL model inference is completed within the function infer_run. Unlike physics-based radiation schemes, DL-based schemes do not need to calculate intermediate varaibles such as optical depth. Instead, the DL-based model accomplishes the mapping from the input features to outputs. After inference is finished, DL model outputs will be post-processed to match the dimensions of the original WRF data array to continue model simulations.





```
subroutine radiation_driver
        defining variables
        initializing data
        call preprocess_layout
        call infer_run
        call postprocess
end subroutine radiation_driver
```

**Figure 2.** Pseudo-code for the DL-based RRTMG module.

## 3   WRF-DL coupler

This subsection briefly describes the WRF I/O quilt server techniques as a similar method is adopted in the WRF-DL coupler to have exclusive servers for DL model inference. Then the methodologies of the WRF-DL coupler are described in detail.

### 3.1   Description of WRF I/O quilt severs

The WRF model is designed to run efficiently on parallel distributed-memory computing systems. To do this, WRF applies domain decomposition to split the entire domain into nprocs (equal to $n_x \times n_y$) number of subdomains so that the total amount of work is divided into nprocs compute tasks, where $n_x$ and $n_y$ are the number of processors along the west-east and south-north directions. Furthermore, the WRF model contains an I/O layer to gather all the distributed data arrays onto the master Message Passing Interface (MPI) rank 0 while other MPI ranks block until the master completes the write. However, as the number of domain sizes increases or resolution increases, leading to an increasing amount of information to write, the increasing time cost of writing output files becomes a bottleneck for the overall performance. Therefore, the WRF model also provides an option to use asynchronous I/O (quilt) servers that deal exclusively with I/O so that the compute processors can continue without waiting for writing output files. More specifically, WRF allows users to specify the number of groups of I/O servers to allocate (nio_groups) and the number of I/O ranks per group (nio_tasks_per_group). Similar to the I/O quilt servers, the WRF-DL coupler also provides the option to use several processors for DL model inference exclusively, of which more details are described in the following subsection. While GPUs are more powerful than CPUs, both CPUs and GPUs are supported for inference. CPUs are cheaper than GPUs and more widely available, and using CPUs for inference can also avoid extra costs due to GPU-CPU data transfer.

### 3.1.1   Description of the WRF-DL coupler

Figure 3 shows that when the WRF model launches the initialization process, the subroutine infer_init is called to initialize the DL model inference services. Similar to the asynchronous I/O servers, several processes are assigned to exclusively work on DL model inference using either GPUs or CPUs. The number of inference processors and the number of inference ranks per





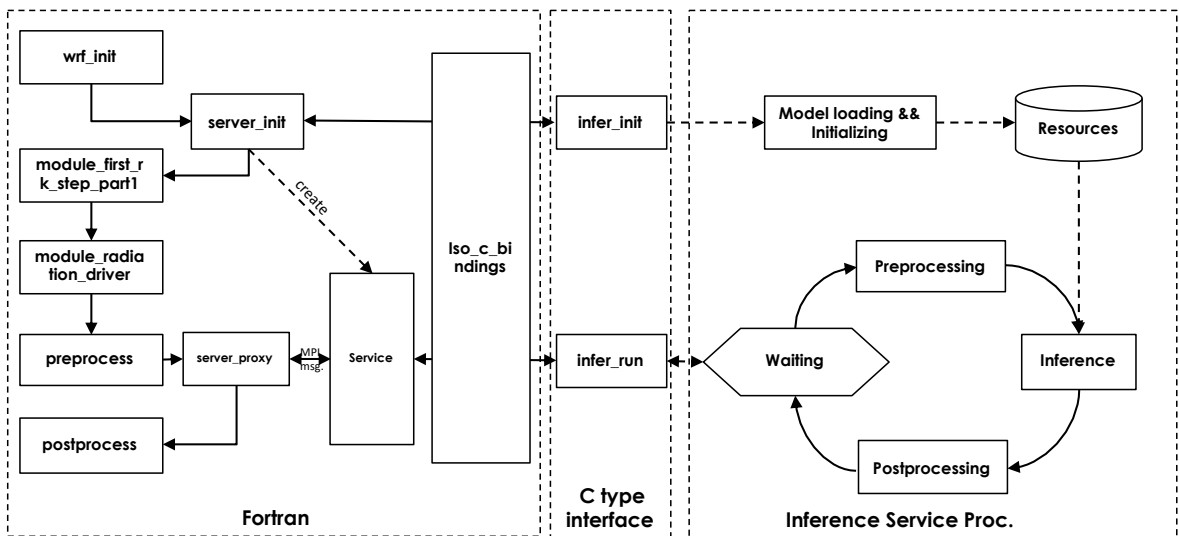

**Figure 3.** Framework of WRF-DL coupler

group are specified in the configuration file by setting the newly added variables ninfer_tasks_per_group and ninfer_groups in the namelist.input file. The difference between the inference processors and the asynchronous I/O servers is that the inference processors apply the synchronous method, as WRF calculations run sequentially, and the subsequent calculations of

the WRF model rely on the outputs from DL-based parameterizations. Usually, the number of inference processors is much smaller than that of WRF compute processors, so a single inference processor must process data from multiple processors. The inference processor listens and receives data from corresponding WRF compute processors through the MPI transmission while blocking the compute processes (see the right part in Figure 4). After inference is finished, the inference process sends outputs of DL-based schemes back to the compute processors to continue the standard WRF calculations. In addition, the

ninfer_tasks_per_group and ninfer_groups can be set to -1 to execute the DL model inference using the same processors for WRF compute processes. Then, memory copy is not needed between WRF compute processors and inference processors.

The functions for implementing the DL-based model inference are all written in Python, commonly used by the machine learning community. To allow the Fortran-based WRF code to call those Python functions, it is necessary first to link the Fortran executables to the system Python library. iso_c_binding is an intrinsic module defined in Fortran 2003 for interoperability with

C. As shown in the left part of Figure 3, the iso_c_binding is imported within the Fortran code. Although no C code is needed, the C Foreign Function Interface (CFFI) for Python is used to generate c-style binding interfaces within the Python scripts (see the middle part of Figure 3). The interfaces for C-function are also defined in the Python scripts, which are used to generate a dynamic library that Fortran modules can link.

This WRF-DL coupler is superior because it makes the DL models easily coupled with the WRF model with minimal effort.

Also, researchers are able to make full use of the state-of-art DL model structures. This methodology is made open-source in the hope that more and more machine learning researchers will participate in improving the traditional NWP models.



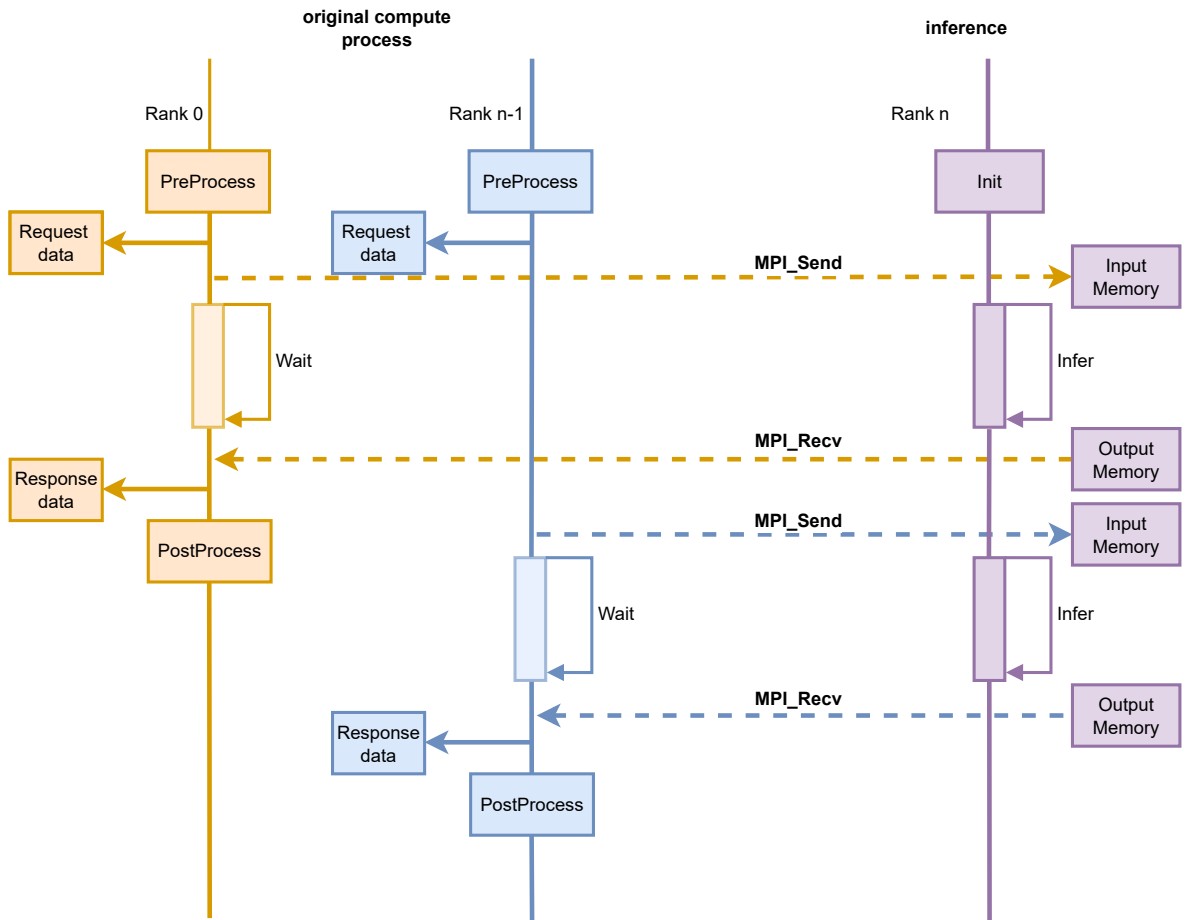

**Figure 4.** Synchronous sequence implementation between WRF and DL-based parameterization

## 4   Experiment Setup

The DL-based radiation emulator is coupled with the WRF model to demonstrate the coupler's practicality. This section firstly describes the WRF model setup, DL-based radiation emulators' network structures, and offline evaluation metrics.

### 4.1   WRF Model Setup

In this work, the WRF model version 4.3 is compiled using the GNU Fortran (gfortran) compiler with the "dmpar" option. To generate a dataset for model training, the WRF model is run using the domain configuration shown in Figure 5. In addition, the WRF model is configured using physics schemes, including New Thompson et al. scheme (Thompson et al., 2008), Bougeault-Lacarrère (BouLac) planetary boundary layer (PBL) scheme (Bougeault and Lacarrère, 1989), RUC land surface model (Smirnova et al., 1997; Smirnova et al., 2000), and RRTMG for both SW and LW radiation (Iacono et al., 2008). The





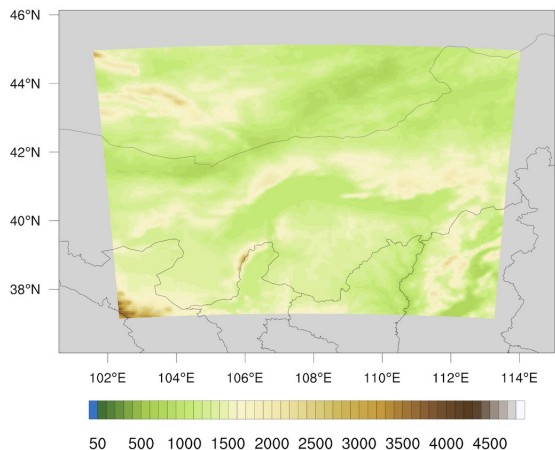

**Figure 5.** Digital evaluation data of the single WRF domain with horizontal resolution at 5 km.

WRF simulations were run for three days per month, and the initialization days were randomly selected. The first two days of every three days' simulations are used for training, and the last day is used for testing. The model generates radiation inputs and outputs every 30 model minutes.

## 4.2 DL-based radiation emulators and offline evaluations

Many researchers (Chevallier et al., 2000; Krasnopolsky et al., 2010; Song and Roh, 2021) have previously used the FC networks to replace the radiation schemes within the operational NWP models. Additionally, Yao et al. (2022) demonstrated that the bidirectional long short-term memory (Bi-LSTM) achieves the best accuracy in emulating radiation. Therefore, this study has trained and tested both FC networks and Bi-LSTM models. As an increasing number of trainable parameters of a DL model increases both accuracy and computational cost, finding a balance is crucial for applying DL models in operational

NWP models. Table 1 illustrates the structures of five DL models, including two FC networks (models A and B) and three Bi-LSTM models (models C to E). Models A and D, as well as models B and E, have approximately the same number of parameters, respectively. More details about the model training and comparisons can be referenced in Yao et al. (2022). All the DL-based emulators have been converted to ONNX and run using the ONNX Runtime library version 1.7.

Figure 6 shows the inference time of different DL-based radiation emulators on one Intel Xeon Platinum 8269CY CPU and

one NVIDIA Tesla T4 GPU, respectively. Model E has the longest inference time and is much longer than Model B, even though they have a similar number of parameters. This is because the FC networks have much lower time complexity than LSTM models. Similarly, although the number of parameters of model A is about 10.3 and 1.03 times that of models C and D (see Table 1), Model A is still faster than models C and D.

The performance of different DL-based radiation emulators is evaluated on the testing data in the offline setting. Table 2

summarizes the RMSE of fluxes and heating rates which are the final outputs of the radiation schemes. Since SW fluxes at





**Table 1.** Description of different DL models used for emulating radiative transfer calculations, including description of model structures and total number of parameters.

| Model Name | Network Structure | Layer Number | Node Number | Parameter Number |
|------------|-------------------|--------------|-------------|------------------|
| Model A | FC | 5 | 30 | 70248 |
| Model B | FC | 10 | 200 | 840028 |
| Model C | Bi-LSTM | 1 | 16 | 6788 |
| Model D | Bi-LSTM | 3 | 32 | 67844 |
| Model E | Bi-LSTM | 5 | 96 | 993028 |

**Table 2.** RMSE of different DL-based radiation emulators for test data in offline evaluation

| | SW Flux $W \cdot m^{-2}$ | LW Flux $W \cdot m^{-2}$ | SW Heating Rate $K \cdot d^{-1}$ | LW Heating Rate $K \cdot d^{-1}$ | Surface SW Flux $W \cdot m^{-2}$ | TOA SW Flux $W \cdot m^{-2}$ |
|---|---|---|---|---|---|---|
| Model A | 2.470e+02 | 4.146e+01 | 1.172e+00 | 2.035e+00 | 2.092e+02 | 2.735e+02 |
| Model B | 1.952e+02 | 2.964e+01 | 9.455e-01 | 2.172e+00 | 1.733e+02 | 2.163e+02 |
| Model C | 7.658e+00 | 2.032e+00 | 2.772e-01 | 3.373e-01 | 8.534e+00 | 6.628e+00 |
| Model D | 5.355e+00 | 1.305e+00 | 9.943e-02 | 1.614e-01 | 5.854e+00 | 4.804e+00 |
| Model E | 6.157e+00 | 1.274e+00 | 5.091e-02 | 9.520e-02 | 6.871e+00 | 5.576e+00 |

the surface and top-of-atmosphere (TOA) radiation is particularly important to climate and weather prediction, their RMSE is also shown in Table 2. It is shown that model E is the most accurate, with RMSE of SW and LW fluxes of about 6.157 and 1.274 $W/m^2$, RMSE of SW and LW heating rates of about $5.091 \times 10^{-2}$ and $9.520 \times 10^{-2}$ $K/day$, and SW fluxes at the surface and TOA of about 6.871 and 5.576 $W/m^2$. Model D has comparable accuracy to the best-performing model E while having only 1/14 of the parameters and a much shorter inference time. Furthermore, considering that model C only has the smallest number of parameters, model C has relatively good accuracy while the RMSE of SW heating rate is five times higher than model E. On the other hand, models A and B have at least four times higher RMSE than model E in terms of heating rates, with RMSE of SW fluxes at the surface and top-of-atmosphere at least 20 times higher than model E. In summary, it is demonstrated that the Bi-LSTM model is more accurate than FC networks for radiation emulation, and model D achieves the best balance between accuracy and computing efficiency.

Since it is difficult to know the performance of the DL-based emulators in the online settings based on offline evaluation, the following Section presents the online evaluations of different DL-based emulators coupled with the WRF model. In addition, the WRF model is also run with the original RRTMG schemes as a reference to evaluate the performance of the DL-based emulators.





**(a)**

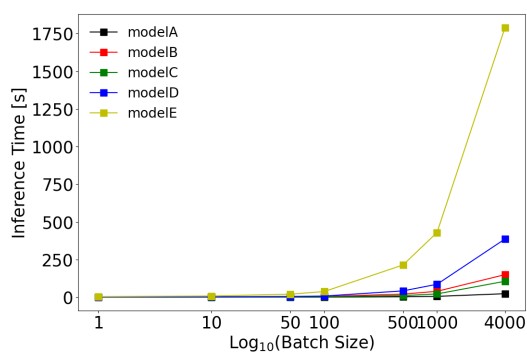

**(b)**

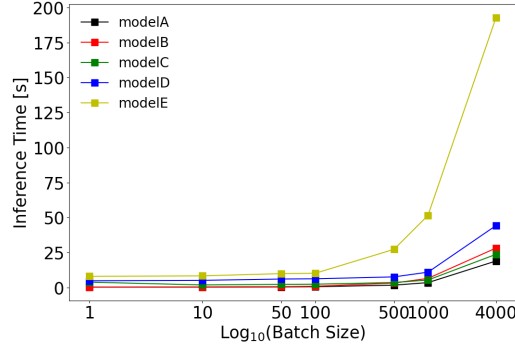

**Figure 6.** Inference time of different DL-based emulators run on one CPU core (6a) and GPU (6b) with different batch size.

## 5 Results

The DL-based radiation emulators are coupled with the WRF model using the WRF-DL coupler. The WRF simulations coupled with Dl-based emulators run using three configurations: zero exclusive processes for inference, four exclusive inference processes on CPUs, and four exclusive inference processes on CPUs. Also, the total number of WRF processes is kept to 24 to ensure fair comparisons. The calculation time of radiation driver using different DL-based radiation emulators is compared. The WRF model is initialized at 12 UTC on a randomly selected day (i.e., November 9th) in the year 2021 and runs for three days, which is not part of the previously used dataset to ensure the DL-based emulators are evaluated on unseen data. Finally, the temperature at 2 meters ($T_{2m}$) and wind speed at 10 meters ($WS_{10m}$) output from the WRF model are compared.

### 5.1 Computational efficiency evaluation of radiation driver

The calculation time of the WRF radiation driver using original RRTMG schemes and DL-based radiation emulators are profiled using the timing function rsl_interal_microclock provided by the WRF model. Table 3 shows that it takes about 1441



**Table 3.** Calculation time (in units of milliseconds) of radiation driver when using the original RRTMG schemes and the DL-based radiation emulators.

| Total Processes | 24 | | |
|---|---|---|---|
| original RRTMG | 1441.24 ms | | |
| DL Inference Processes | 0 | 4 CPU | 4 GPU |
| Model A | 91.73 | 237.57 | 176.87 |
| Model B | 365.31 | 808.43 | 331.34 |
| Model C | 666.37 | 921.56 | 258.66 |
| Model D | 2620.92 | 3383.42 | 418.28 |
| Model E | 7166.46 | 10330.05 | 1690.76 |

milliseconds to run a radiation driver using the original RRTMG schemes. In terms of DL-based radiation emulators, using GPUs for inference is significantly faster than using CPUs for inference, especially for the Bi-LSTM model, as it is less computationally efficient than the FC networks given a similar number of parameters. When GPUs are used for inference, all DL-based emulators except model E are at least three times faster than the original RRTMG schemes. On the other hand, when
CPUs are used for inference, having four processes exclusively for inference is slower than having inference executed on the WRF compute processors, which is probably due to the more significant increase in time cost due to data copy between CPUs than the decrease in time cost from having CPUs exclusively for inference.

## 5.2 Accuracy evaluation

Figures 7 and 8 present the difference in 24, 48, and 72 hours forecast between the WRF simulations coupled with DL-
based radiation emulators and the WRF simulation with original RRTMG schemes. On the spatial difference, the red and blue patterns indicate significantly positive and negative biases of DL-based simulations, respectively. On the other hand, green patterns indicate no or small difference from the original WRF simulations. A domain averaged mean absolute difference (MAD) is also calculated to measure the overall performance of DL-based emulators in online settings. Notably, the difference does not increase with the simulation time, as the difference at 72 forecast hours is similar to that of 24 forecast hours. The
WRF simulations coupled with FC networks (models A and B) have the worst performance, with a much larger difference from the original WRF simulation than Bi-LSTM models over the entire domain. Model B is more accurate than model A in offline settings but performs worse than model A in the online evaluation. The MAD of $T_{2m}$ and $WS_{10m}$ is greater than 2.3 and 1.0, respectively, for model B, while the MAD of $T_{2m}$ and $WS_{10m}$ is slightly less than 3 and 1.0, respectively, for model A. Model C. Models D and E have a similar spatial distribution of difference, although model E has about ten times
the number of parameters as model D and slightly higher accuracy in offline validation. From the comparisons of offline and online performance of Dl-based parameterization, it can be concluded that the DL-based parameterization's offline skills can not reflect their online skills. Therefore, it is necessary to have DL-based parameterization coupled with the NWP models and evaluated in online settings.



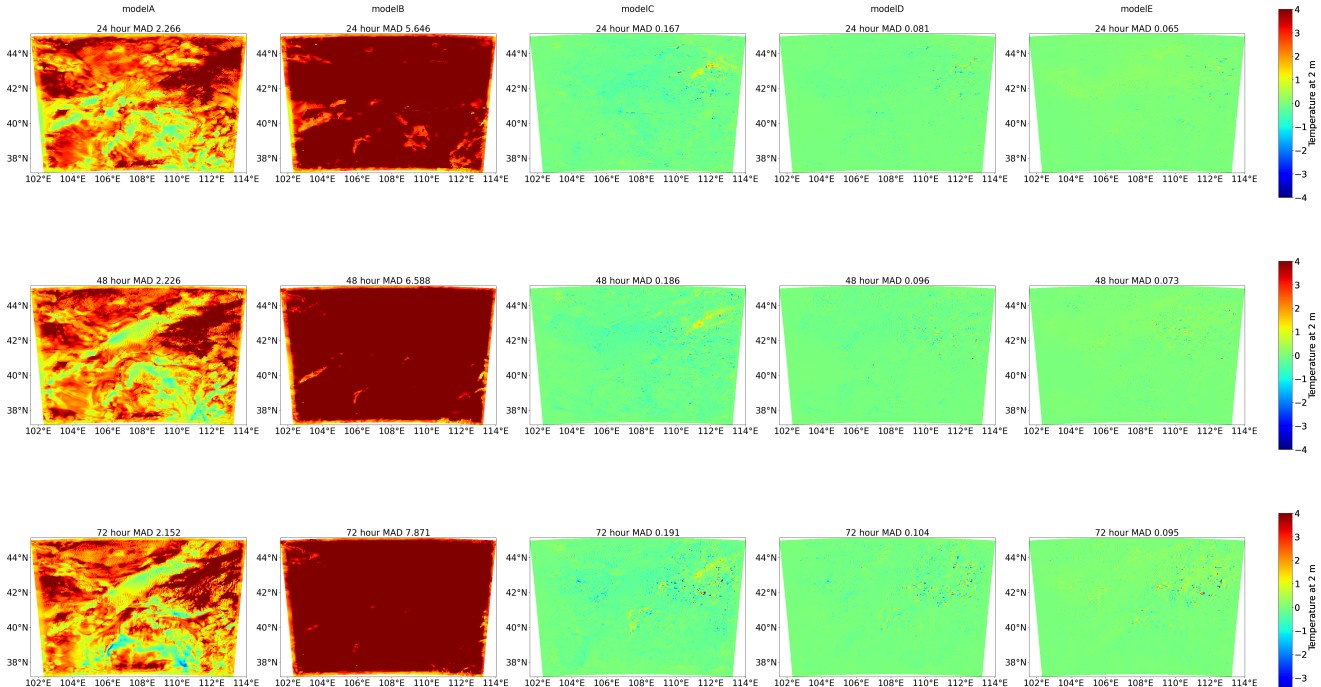

**Figure 7.** Differences in temperature at 2 meters at forecast horizons of 24, 48, and 72 hours between WRF simulations coupled with DL models A, B, C, D, and E and WRF simulation with the original RRTMG schemes for radiation emulation. The title of each figure indicates the mean absolute difference (MAD) of the WRF domain.

Overall, model E demonstrates the best performance, with the domain averaged MAD smaller than 0.10 for both temperature
at 2 meters and wind speed at 10 meters. Both model D and model E show a comparable forecast with the original WRF simulation over the entire domain. Overall, model D is more suitable to replace the original RRTMG schemes as it is more computationally efficient than model E. Using model D with GPU as inference processors is about 7 times faster.

In summary, the WRF-DL coupler enables all the DL-based radiation emulators coupled with the WRF models more efficiently. Moreover, the WRF simulations coupled with DL models are run successfully using CPU or GPU for inference.
Furthermore, the WRF-DL coupler can be a valuable tool for researchers to evaluate their DL-based parameterization in online settings.





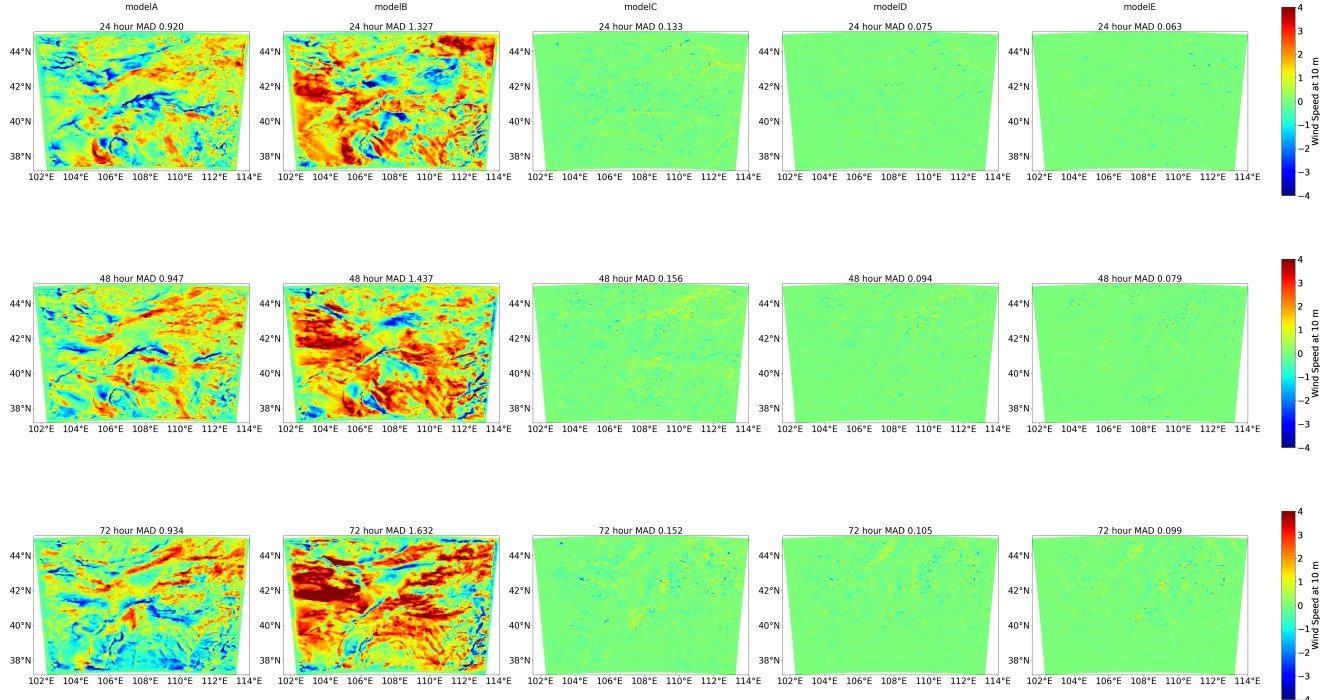

**Figure 8.** Differences in wind speed at 10 meters at forecast horizons of 24, 48, and 72 hours between WRF simulations coupled with DL models A, B, C, D, and E and WRF simulation with the original RRTMG schemes for radiation emulation.

## 6   Summary and Conclusions

As demonstrated in many previous studies, emulating the subgrid physics using DL models has the potential to be faster and even more accurate than the conventional physics-based parameterization schemes. However, the fact that DL models

are commonly implemented using Python while the NWP models are coded in Fortran makes it challenging to implement the DL-based parameterization in NWP models for operational applications. Previously, researchers usually hard coded the operations of NNs into Fortran, which is time-consuming and prone to error as even minor changes to the DL model require rewriting the Fortran code. Some researchers used the Fortran-based NN libraries to access features of the Python-based deep learning frameworks in Fortran. However, these methods are only feasible for applying simple, dense, layer-based NNs. In this

paper, we propose a WRF-DL coupler that provides researchers with the tool to implement DL models into NWP models with minimal effort. The advantage of this coupler is that more complex and advanced art DL model structures are supported as the implementation of DL-based parameterization is achieved in Python. The coupler is made open source in the hope that more and more researchers can integrate the DL-based parametrization to improve the traditional NWP models for more accurate weather or climate forecast.



Furthermore, the capability of the WRF-DL coupler was demonstrated by coupling the DL-based radiation parameterization into the WRF model. It was also illustrated that DL-based parameterizations' computational efficiency and accuracy should be balanced to achieve the best overall performance for operational applications.

*Code and data availability.* The source code and data used in this are available at https://doi.org/10.5281/zenodo.7056166 (Zhong et al., 2022)

*Author contributions.* Y.Y. trained the deep learning models and calculated the statistics of models' offline performance. X.Z., Y.W., and Z.M. modified the WRF source code and developed the WRF-DL coupler. X.Z. conducted the WRF model simulations coupled with the original RRTMG schemes for DL model training and offline evaluation. X.Z. and Z.M. conducted the WRF simulations coupled with DL-based radiation emulators. X.Z. wrote, reviewed, and edited the original draft; Z.W. supervised and supported this research and gave valuable opinions. All of the authors have contributed to and agreed to the published version of the manuscript.

*Competing interests.* The authors declare no conflict of interest.

*Acknowledgements.* This work was supported in part by the Zhejiang Science and Technology Program under Grant 2021C01017.



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
