# Peer review of "WRF-ML v1.0: A Bridge between WRF v4.3 and Machine Learning Parameterizations and its Application to Atmospheric Radiative Transfer"

_EGUsphere, 2022_

## Referee Comment (RC1)

**Review for Zhong et al., 2022, WRF-DL v1.0: A Bridge between WRF v4.3 and Deep Learning Parameterizations and its Application to Atmospheric Radiative Transfer**

The manuscript presents the new WRF-DL coupler, which allows the integration of the machine learning models, written in Python, into the WRF weather forecasting model, written in Fortran. In general, a lot of modern software is written in Python, while the bulk of the models in geosciences is written in Fortran due to demands in computationally efficiency. Integration of the Python written modules into the Fortran codes is often desired but hampered by the lack of the appropriate interfacing infrastructure. And, although the authors focus on deep learning only, the methodology presented in this manuscript is general and allows integration of any Python models (including physical, i.e. Mie calculation or cloud parametrizations, etc) into WRF. A further advantage of this work is that it presents a functional example of Python-Fortran coupling and can serve as a template in the future.

The topic of the manuscript aligns well with the focus of the journal and the work represents important progress in model development capabilities. The manuscript is very well written and I enjoyed reading it. I recommend publishing this work after some of the non-major comments are addressed.

**Minor comments:**

The use of the term "Accuracy" in section 5.2 is inappropriate. Accuracy should refer to the comparison of the models output, which are driven by the same input. It is appropriate to use the term "accuracy" for the model comparison in the offline regime. The online regime, however, does not permit such comparison because the input parameters evolve in time and are not consistent between different runs. By propagating small differences in time, the model transitions into a different physical state, which is characterized by the individual temperature and velocity profile.

The fundamentals of this issue were studied by Lorenz and are known as the "weather predictability limit" (https://doi.org/10.3402/tellusa.v34i6.10836).

To summarize, It is OK to keep section 5.2, but the wording needs to be adjusted and the conclusion on the 220-222 is trivial.

Line 193. CPUs->GPUs
Line 200. rsl_interal_microclock -> rsl_inteNral_microclock
Line 229. More efficiently than what?

Finally, I encourage offers to prepare and document a minimal functional example of coupling the primitive python subroutine into the WRF code (in addition to what is already provided as a git repository). This request is beyond the scope of the manuscript and elective.

---

## Referee Comment (RC2)

**Review for Zhong et al., 2022: *WRF-DL v1.0: A Bridge between WRF v4.3 and Deep Learning Parameterizations and its Application to Atmospheric Radiative Transfer**

This is a fairly well-written and pleasantly concise paper on a new module developed for WRF which, as I understand, can be used to communicate with any existing ML/DL libraries written in or accessible from Python, and easily implement ML-based parameterizations in WRF. While I lack the expertise to comment on computational and programming aspects of the coupler, I think it's clearly a good idea. The demonstration with a radiation emulator is relevant and useful. I have three more important points which should be addressed:

1. Before reading the review by RC1 I did not understand that the coupler can be used to communicate with any Python script, and not a specific DL library. For the non-technical reader like myself the basic principles, and how a user would go about to use this coupler in practice, should probably be clarified in Section 3.

2. If you are hoping that other people used this tool, should it not be added to a proper code repository like Github where it can be maintained?

3. I agree with RC1 about the evaluation in 5.2 being problematic, especially if you call it "accuracy". If you are measuring anything here, it is more likely "generalization" (it's not surprising that the complex models perform worse, they are overfitted), but it is altogether not particularly useful to compare a couple realizations of instantaneous fields. However, since the topic of the paper is the coupler and not the emulator, I think it's OK to keep this basic online evaluation if the limitations are mentioned.

Besides this, I only have minor comments, mostly concerning clarity and grammar, which are easily addressed. After the authors have done so, I recommend that the paper is published.

**Minor comments**

**L69. *"DL-based radiation emulators were studied most..."***
Technically the emulators in most of these papers were not DL, as they were based on shallow neural networks. You could change to "ML-based", or perhaps just leave it as it is and choose simplicity over correctness

**L72: *"...such as ECMWF IFS"***
RRTMG, the full radiation scheme you used in your study, is not used in the IFS. IFS uses ecRad (Hogan & Bozzo 2018), which is a modular radiation scheme that has the gas optics component from RRTMG, but is otherwise completely different. You could either not mention the IFS or you could add, "and its gas optics component is also used in the ECMWF IFS".

**L99: *"three-dimensional variables are converted to be indexed a (i,j,k)"***
This sentence is confusing, are you converting from 3D to 3D?

**L120: *"While GPUs are more powerful than CPU's"***
In my own tests, GPU's have not necessarily been faster for small networks, which might be used for parameterizations. You could instead write *"While GPUs are typically more powerful than CPUs for DL inference, both CPUs and GPUs are supported"*.

**L135: *"..for WRF compute processes"* → *"..as for WRF compute processes"***

**L144. *"This WRF-DL coupler is superior because.."***
Superior to what?

**L151. The compiler version could be added.**

**L160-163.** *"Yayo et al. (2022) demonstrated that the bidirectional..."*
As far as I'm aware (the Yao study seems to be a preprint?), my own paper (Ukkonen, 2022) was the first to use bidirectional RNNs for radiative transfer and demonstrate they're more accurate than FNNs, so the paper should be mentioned here

**Table 1:** Does "layer number" refer to the number of bidirectional layers, meaning model E actually does 10 traversals through the vertical column (2*5)? I am asking out of interest, perhaps it need not be clarified in the paper.

**Figure 6:** You should mention the problem size (number of columns and their vertical resolution) in the caption.

**L192.** *"emulators run"* → *"emulators are run"* / *"emulators were run"*

**L193. End of sentence:** *"CPUs"* → *"GPUs"?*

**Table 3:** Problem size should be mentioned. I think all emulator studies should report the time per column in one way or another, and the hardware, so we can compare results between studies: it is a problem in many studies that only the speed-up is reported (but this is always relative, the original code can be slow)

**L203.** *"When GPUs are used for inference, all DL-based emulators except model E are at least three times faster than the original RRTMG scheme"*
But what if you had used the same dedicated GPUs for RRTMG, would it still be faster? This comparison has some caveats which should be mentioned. It is unclear if it's the software (ML model) that is faster or the hardware.

**L219:** *"model A. Model C".* → *"Model A and Model C"?*

**L233-235:** This statement could be clarified a little bit. For instance:
*"..has the potential to be faster, or by training on observations or detailed high-resolution models, even more accurate than conventional parameterizations"*

**L238.** *"Some researchers..."* This sentence is confusing. If you are referring to Fortran-Keras bridge, perhaps you mean:
*"Some researchers used simple Fortran-based NN libraries, which could convert existing models trained in Python to models usable in the Fortran framework."*

**L241**. Remove "art"

**L242**. *"..as the implementation of DL-based parameterization is achieved in Python"*
This sentence is a bit convoluted. *"...as it can communicate with Python-based frameworks"?*

**References**

**Hogan, R. J. and Bozzo,** A.: A flexible and efficient radiation scheme for the ECMWF model, Journal of Advances in Modeling Earth Systems, 10, 1990–2008, https://doi.org/https://doi.org/10.1029/2018MS001364, **2018**

**Ukkonen, P.**: Exploring pathways to more accurate machine learning emulation of atmospheric radiative transfer, Journal of Advances in Modeling Earth Systems, p. e2021MS002875, https://doi.org/10.1029/2021MS002875, **2022**

---

## Author Comment (AC1)

**Reply to reviewers' comments**

We thank the reviewer for the time spent on reviewing this manuscript and for providing helpful comments and suggestions.

**Reviewer #1**

The manuscript presents the new WRF-DL coupler, which allows the integration of the machine learning models, written in Python, into the WRF weather forecasting model, written in Fortran. In general, a lot of modern software is written in Python, while the bulk of the models in geosciences is written in Fortran due to demands in computationally efficiency. Integration of the Python written modules into the Fortran codes is often desired but hampered by the lack of the appropriate interfacing infrastructure. And, although the authors focus on deep learning only, the methodology presented in this manuscript is general and allows integration of any Python models (including physical, i.e. Mie calculation or cloud parametrizations, etc) into WRF. A further advantage of this work is that it presents a functional example of Python-Fortran coupling and can serve as a template in the future. The topic of the manuscript aligns well with the focus of the journal and the work represents important progress in model development capabilities. The manuscript is very well written and I enjoyed reading it. I recommend publishing this work after some of the non-major comments are addressed.

**Response: We thank the reviewer for their comments and suggestions to improve the manuscript. We added the "In addition, the methodologies also allow the integration of any Python script into the WRF model. An example of coupling the primitive Python subroutines into the WRF code is also provided in the code repository." in the Section 6.**

**Minor comments:**

The use of the term "Accuracy" in section 5.2 is inappropriate. Accuracy should refer to the comparison of the models output, which are driven by the same input. It is appropriate to use the term "accuracy" for the model comparison in the offline regime. The online regime, however, does not permit such comparison because the input parameters evolve in time and are not consistent between different runs. By propagating small differences in time, the model transitions into a different physical state, which is characterized by the individual temperature and velocity profile.

The fundamentals of this issue were studied by Lorenz and are known as the "weather predictability limit" (https://doi.org/10.3402/tellusa.v34i6.10836)

To summarize, It is OK to keep section 5.2, but the wording needs to be adjusted and the conclusion on the 220-222 is trivial.

**Response: We agree that the term "Accuracy" in subsection 5.2 is not appropriate as the comparisons were between the WRF simulations coupled with ML-based radiation emulators and the WRF simulation with original RRTMG schemes instead of observations. We changed the subsection title to "Quantitative evaluation". The conclusions on the line 220-222 are removed.**

Line 193. CPUs->GPUs

**Response: Changed to "GPUs".**

Line 200. rsl_interal_microclock -> rsl_inteNral_microclock

**Response: Changed to "rsl_internal_microclock"**

Line 229. More efficiently than what?

**Response: Changed to "efficiently".**

Finally, I encourage offers to prepare and document a minimal functional example of coupling the primitive python subroutine into the WRF code (in addition to what is already provided as a git repository). This request is beyond the scope of the manuscript and elective.

**Response: As suggested by the reviewer, we add and document an example of coupling the primitive Python subroutine into the WRF code in the code repository.**

**Reviewer #2**

This is a fairly well-written and pleasantly concise paper on a new module developed for WRF which, as I understand, can be used to communicate with any existing ML/DL libraries written in or accessible from Python, and easily implement ML-based parameterizations in WRF. While I lack the expertise to comment on computational and programming aspects of the coupler, I think it's clearly a good idea. The demonstration with a radiation emulator is relevant and useful. I have three more important points which should be addressed:

**Response: We thank the reviewer for their comments and suggestions to improve the manuscript. We address the concerns below.**

1.Before reading the review by RC1 I did not understand that the coupler can be used to communicate with any Python script, and not a specific DL library. For the non-technical reader like myself the basic principles, and how a user would go about to use this coupler in practice, should probably be clarified in Section 3.

**Response: We added the following sentences in Section 6 to inform readers that the coupler can be used to communicate with any Python script: "In addition, the methodologies also allow the integration of any Python script into the WRF model. An example of coupling the primitive Python subroutines into the WRF code is also provided in the code repository.". Also, we added instructions on how to use this coupler in the ReadMe file in the GitHub repository (https://github.com/x7zhong/WRF-DL).**

2.If you are hoping that other people used this tool, should it not be added to a proper code repository like Github where it can be maintained?

**Response: The code is available at https://doi.org/10.5281/zenodo.7056166 and has also been uploaded to GitHub (https://github.com/x7zhong/WRF-DL).**

3. I agree with RC1 about the evaluation in 5.2 being problematic, especially if you call it "accuracy". If you are measuring anything here, it is more likely "generalization" (it's not surprising that the complex models perform worse, they are overfitted), but it is altogether not particularly useful to compare a couple realizations of instantaneous fields. However, since the topic of the paper is the coupler and not the emulator, I think it's OK to keep this basic online evaluation if the limitations are mentioned.

**Response: We agree that the term "Accuracy" in subsection 5.2 is not appropriate as the comparisons were between the WRF simulations coupled with ML-based radiation emulators and the WRF simulation with original RRTMG**

**schemes instead of observations. Therefore, we changed the title of subsection 5.2 to "Quantitative evaluation".**

**Minor comments**

**L69. "DL-based radiation emulators were studied most..."**
Technically the emulators in most of these papers were not DL, as they were based on shallow neural networks. You could change to "ML-based", or perhaps just leave it as it is and choose simplicity over correctness

**Response: We agree that a lot of previous studies applied ML models. Since deep learning is a subset of machine learning, we changed all the appearances of "deep learning" and "DL" to "machine learning" and "ML".**

**L72: "...such as ECMWF IFS"**
RRTMG, the full radiation scheme you used in your study, is not used in the IFS. IFS uses ecRad (Hogan & Bozzo 2018), which is a modular radiation scheme that has the gas optics component from RRTMG, but is otherwise completely different. You could either not mention the IFS or you could add, "and its gas optics component is also used in the ECMWF IFS".

**Response: We mentioned ECMWF IFS as it is mentioned on this website http://rtweb.aer.com/rrtm_frame.html. We agree that the radiation scheme used in the current operational ECMWF IFS is ecRad, so we removed "ECMWF IFS" from the sentence.**

**L99: "three-dimensional variables are converted to be indexed a (i,j,k)"**
This sentence is confusing, are you converting from 3D to 3D?

**Response: There was a typo in the sentence "are indexed as (i, j, k) in memory by WRF to make it more cache friendly and increase the cache hit rate." Moreover, we change from "(i, j, k) in memory" to "(i, k, j) in memory". By WRF coding conventions, the three-dimensional variables are indexed as (i, k, j). Therefore, we reshape the variables from being indexed as (i, k, j) to (i, j, k). We also changed the wording from "are converted" to "are reshaped" to clarify it.**

**L120: "While GPUs are more powerful than CPU's"**
In my own tests, GPU's have not necessarily been faster for small networks, which might be used for parameterizations. You could instead write "While GPUs are typically more powerful than CPUs for DL inference, both CPUs and GPUs are supported".

**Response: We agree with the reviewer that using GPUs for inference is not necessarily faster than using CPUs. So, we rewrote the sentence as the reviewer suggested.**

**L135: "..for WRF compute processes" → "..as for WRF compute processes"**

**Response: Changed to "… as for WRF compute processes"**

**L144. "This WRF-DL coupler is superior because.."**
Superior to what?

**Response: We changed the sentence from "This WRF-DL coupler is superior because …" to "This WRF-ML coupler allows the ML models to be easily coupled with …".**

L151. The compiler version could be added.

**Response: The compiler version is added as "gfortran version 6.5.1"**

**L160-163. "Yayo et al. (2022) demonstrated that the bidirectional..."**
As far as I'm aware (the Yao study seems to be a preprint?), my own paper (Ukkonen, 2022) was the first to use bidirectional RNNs for radiative transfer and demonstrate they're more accurate than FNNs, so the paper should be mentioned here

**Table 1:** Does "layer number" refer to the number of bidirectional layers, meaning model E actually does 10 traversals through the vertical column (2*5)? I am asking out of interest, perhaps it need not be clarified in the paper.

**Figure 6:** You should mention the problem size (number of columns and their vertical resolution) in the caption.

**Response: The reviewer mentioned is the first paper to apply bidirectional RNNs for emulating atmospheric radiative transfer. We added the following sentence in Subsection 4.2: "Additionally, Ukkonen (2022) showed that bidirectional recurrent NNs (RNNs) are more accurate than the feed-forward NNs (FNNs) for emulating atmospheric radiative transfer."**

**"Layer Number" in Table 1 refers to the number of bidirectional layers if the model is a Bi-LSTM model. Moreover, we clarify it by changing the "Layer Number" to "Number of Layers (Bidirectional Layers for Bi-LSTM models)".**

**We added "The number of vertical levels is 57." in the caption of Figure 6. The vertical resolution does not affect the ML model inference time here. We also added more details of the WRF model setup in Subsection 4.1: "The WRF model**

is set up within the GFS model grid with a single domain at a horizontal grid spacing of 5 km and has 190 by 170 west-east and north-south grid points (i.e., 32300 grid cells). The number of vertical levels is 57, and the model top is 10 hPa.".

L192. "emulators run" → "emulators are run" / "emulators were run"

Response: Changed to "emulators were run"

L193. End of sentence:"CPUs" → "GPUs"?

Response: Changed to "GPUs"

Table 3: Problem size should be mentioned. I think all emulator studies should report the time per column in one way or another, and the hardware, so we can compare results between studies: it is a problem in many studies that only the speed-up is reported (but this is always relative, the original code can be slow)

Response: The problem size has been added in the caption of Table 3: "The batch size is 32300, as the domain has 190 × 170 grid points."

L203. "When GPUs are used for inference, all DL-based emulators except model E are at least three times faster than the original RRTMG scheme"
But what if you had used the same dedicated GPUs for RRTMG, would it still be faster? This comparison has some caveats which should be mentioned. It is unclear if it's the software (DL model) that is faster or the hardware.

Response: We agree that GPUs could also possibly accelerate the original RRTMG schemes. So we added: "However, whether this speed-up is due to applying DL-based emulators or the upgrade in hardware (using CPUs instead of GPUs) is unclear, which is beyond the scope of this paper.". This work's focus is on the coupler, and the results part is a basic evaluation.

L219: "model A. Model C". → "Model A and Model C"?

Response: Changed to "model A"

L233-235: This statement could be clarified a little bit. For instance:

"..has the potential to be faster, or by training on observations or detailed high-resolution models, even more accurate than conventional parameterizations"

**Response: Changed to "has the potential to be faster, or by training on observations or detailed high-resolution models, even more accurate than the conventional parameterization schemes."**

**L238. "Some researchers..."** This sentence is confusing. If you are referring to Fortran-Keras bridge, perhaps you mean:
"Some researchers used simple Fortran-based NN libraries, which could convert existing models trained in Python to models usable in the Fortran framework."

**Response: Changed to "Some researchers used s Fortran-based NN libraries, which could convert existing models trained in Python to models usable in the Fortran framework."**

**L241.** Remove "art"

**Response: Removed.**

**L242.** "..as the implementation of DL-based parameterization is achieved in Python"

This sentence is a bit convoluted. "...as it can communicate with Python-based frameworks"?

**Response: Changed to "as it can directly communicate with Python-based modules.".**

---

## Author Response (AR2)

Comment 1: Thank you very much for revising the manuscript and addressing all referee comments. As a last step before publication, I would like to ask you to update the zenodo repository to make it consistent with the latest git version (including the additional documentation and examples) and to make sure that the link in the "Code and data availability" section points to the updated version.

**Author's response:** The zenodo repository has been updated to be consistent with the latest git version including additional documentation and examples. The updated zenodo repository is https://doi.org/10.5281/zenodo.7407487.

Comment 2: Line 101: "varaibles" -> "variables"

**Author's response:** The typo has been fixed: "varaibles" changed to "variables".

Comment 3: Line 209: "CPUs instead of GPUs" -> "GPUs instead of CPUs"

**Author's response:** The typo has been fixed: "CPUs instead of GPUs" changed to "GPUs instead of CPUs".

Comment 4: Line 232: "... emulators coupled with the WRF models efficiently."
-> "... emulators to couple with the WRF model efficiently." or similar.

**Author's response:** As suggested by the reviewer, the sentence has been changed to "... emulators to couple with the WRF model efficiently."